# A New Wave of Targeting ‘Undruggable’ Wnt Signaling for Cancer Therapy: Challenges and Opportunities

**DOI:** 10.3390/cells12081110

**Published:** 2023-04-08

**Authors:** Woo-Jung Park, Moon Jong Kim

**Affiliations:** 1Department of Life Science, Gachon University, Seongnam 13120, Republic of Korea; 2Department of Health Sciences and Technology, GAIHST, Lee Gil Ya Cancer and Diabetes Institute, Incheon 21999, Republic of Korea

**Keywords:** Wnt signaling, β-catenin, targeted cancer therapy, PROTAC, antibody–drug conjugate (ADC), anti-sense oligonucleotide (ASO)

## Abstract

Aberrant Wnt signaling activation is frequently observed in many cancers. The mutation acquisition of Wnt signaling leads to tumorigenesis, whereas the inhibition of Wnt signaling robustly suppresses tumor development in various in vivo models. Based on the excellent preclinical effect of targeting Wnt signaling, over the past 40 years, numerous Wnt-targeted therapies have been investigated for cancer treatment. However, Wnt signaling-targeting drugs are still not clinically available. A major obstacle to Wnt targeting is the concomitant side effects during treatment due to the pleiotropic role of Wnt signaling in development, tissue homeostasis, and stem cells. Additionally, the complexity of the Wnt signaling cascades across different cancer contexts hinders the development of optimized targeted therapies. Although the therapeutic targeting of Wnt signaling remains challenging, alternative strategies have been continuously developed alongside technological advances. In this review, we give an overview of current Wnt targeting strategies and discuss recent promising trials that have the potential to be clinically realized based on their mechanism of action. Furthermore, we highlight new waves of Wnt targeting that combine recently developed technologies such as PROTAC/molecular glue, antibody–drug conjugates (ADC), and anti-sense oligonucleotides (ASO), which may provide us with new opportunities to target ‘undruggable’ Wnt signaling.

## 1. Wnt Signaling Pathway

In the early 1980s, Wnt signaling was first discovered when *wingless*, an essential developmental gene of Drosophila, was found [1]. Concurrently, *Int-1*, a mammalian wingless homolog, was identified as a driver gene for malignancy when its transcriptional activation was induced by the murine mammary tumor virus [2]. Through *wingless* and *Int-1*, a portmanteau name of ‘Wnt’ was created. Over the four decades since its discovery, Wnt-mediated signaling has been extensively studied, revealing its evolutionarily conserved roles in regulating diverse cellular processes, including embryonic development, tissue homeostasis, and cell fate determination [3].

Wnt signaling consists of Wnt ligands, frizzled receptors (FZD family), co-receptors, β-catenin destruction complexes, β-catenin/transcriptional partners, and other modulating components [3]. Wnt is a secreted ligand and mediates autocrine and paracrine signal transduction through its receptors and downstream effectors [4]. The intracellular downstream signaling of Wnt ligands/receptors is broadly divided into β-catenin-dependent signaling (also referred to as canonical Wnt signaling) and β-catenin-independent signaling (also referred to as non-canonical Wnt signaling) according to its dependence on β-catenin, a central effector of Wnt signaling (Figure 1). The type of Wnt ligands and their corresponding receptors/co-receptors in β-catenin-dependent and -independent Wnt signaling vary in each physiological context. For instance, among the 19 human Wnt ligands, Wnt3a is mainly involved in the β-catenin-dependent Wnt pathway. On the other hand, Wnt5a is predominantly associated with the β-catenin-independent Wnt pathway. However, in some cases, Wnt5A often acts the opposite way [5]. Of human frizzled (FZD) receptors, FZD1 and FZD7 are mainly related to the canonical Wnt pathway. In contrast, FZD2 and FZD6 are implicated with the non-canonical Wnt pathway [5]. However, due to the redundancy and complexity of Wnt signaling, the precise pairs of working Wnt ligands/receptors are still elusive in many biological contexts.

Herein, we briefly describe Wnt signaling transduction, the understanding of which plays a key role in the development of targeting strategies (Figure 1).

### 1.1. Wnt Ligands and Receptors

So far, 19 secreted Wnt ligands and more than 18 Wnt receptors/co-receptors have been identified in the mammalian system [4]. Different combinations of Wnt ligands and their receptors/co-receptors operate in various physiological contexts [6]. Wnt ligands are lipid-modified glycoproteins. The extracellular transport of Wnt requires palmitoylation, a lipid modification mediated by a protein-serine O-palmitoyltransferase, porcupine (PORCN) [7]. Palmitoylated Wnt ligands bind to Wntless and are transported from the Golgi apparatus to the cell membrane for secretion. Secreted Wnts are recognized by the FZD receptor family [3,8]. These G-protein coupled FZD receptors act as primary Wnt receptors and transduce the Wnt signaling intracellularly [8,9]. There are ten FZD receptors in humans. In addition, low-density lipoprotein receptor-related proteins 5 and 6 (LRP5/6) bind FZD and act as typical co-receptors [10]. Additionally, diverse families of Wnt signaling modulators exist. Ring finger protein 43 (RNF43) or zinc and ring finger 3 (ZNFR3) are transmembrane E3 ligases that act as negative regulators of Wnt signaling by inducing the lysosomal degradation of FZD receptors. Conversely, secreted ligands, R-spondins (RSPOs 1-4), serve as Wnt signaling enhancers [11,12]. R-spondins bind to the complex of leucine-rich repeat-containing G protein-coupled receptor 4-6 (LGR4-6) and RNF43/ZNFR3. This interaction prevents the RNF43/ZNFR3-mediated lysosomal degradation of FZD and maintains Wnt signaling transduction [13]. The non-Wnt protein, Norrie Disease Protein (Norrin), also triggers the β-catenin-dependent Wnt pathway [14,15]. Though Norrin is a cystine-knot-like growth factor, it activates Wnt signaling by interacting with FZD4, Lrp5/6, and Tetraspanin-12 [14,15]. Several Wnts, receptors, and co-receptors, and various Wnt modulators operate convergently and divergently. It is also possible that additional unidentified Wnt co-receptors and modulators exist. The complexity of the Wnt signaling cascade makes it challenging to understand the exact responses elicited by the specific targeting of the Wnt signaling steps, which is one of the major difficulties in developing Wnt signaling targeting strategies.

### 1.2. β-Catenin-Dependent Signaling

The β-catenin-dependent Wnt signaling pathway is more elucidated and established compared to the β-catenin-independent pathway (Figure 1). β-catenin is an armadillo repeat protein associated with the cytoplasmic domain of cadherins, cell–cell junction proteins [16,17]. Intracellular levels of non-E-cadherin-bound cytoplasmic β-catenin are very low without Wnt ligand stimulation [18]. The β-catenin destruction complex tightly controls levels of cytoplasm β-catenin. The destruction complex consists of axis inhibitor (AXIN), adenomatous polyposis coli (APC), casein kinase 1 (CK1), glycogen synthase kinase 3β (GSK3β), and β-transducin repeat-containing protein (βTrCP) [10]. In the absence of a Wnt ligand (Wnt-off status), the GSK3β and CK1 of the destruction complex phosphorylate β-catenin at Ser45 (CK1) and Thr41/Ser37/Ser33 (GSK3β) residues, respectively. Phosphorylated β-catenin is sequentially ubiquitinated by the E3 ligase βTrCP and degraded by a ubiquitin-mediated proteasomal system [3]. Upon Wnt ligand stimulation (Wnt-on status), AXIN binds to the phosphorylated cytoplasmic tail of LRP5/6 and the FZD receptor adapter Disheveled (Dvl) [3,8]. This protein interaction sequesters the β-catenin destruction complex from the cytosol to the cell membrane and prevents the destruction complex-mediated degradation of β-catenin. Accumulated β-catenin in the cytoplasm translocates to the nucleus and forms a complex with the transcription factor T cell factor (TCF) or lymphoid enhancer-binding factor 1 (LEF1), replacing Groucho and CtBP corepressors [8,19,20]. β-catenin/transcription factor complexes bind to a co-activator (CBP, p300) for the transcriptional activation of a variety of downstream genes, including c-Myc, Axin2, CCND1, and CD44 [20,21].

### 1.3. β-Catenin-Independent Signaling

In β-catenin-independent signaling, the intracellular downstream signaling of the Wnt/receptor interactions is mediated not by β-catenin but by various cellular signaling modules, such as RAC1-JNK, RHOA-ROCK, and PLC-IP3-Ca^2+^ (Figure 1). Well-characterized physiological contexts of β-catenin-independent signaling include the planar cell polarity (PCP) pathway and Wnt/Ca^2+^ signaling [19]. In Wnt/PCP signaling, binding Wnt ligands to FZDs activates the small GTPases RHOA and RAC1 [22]. Consequently, these signals trigger the activation of Rho-associated protein kinase (ROCK) and JUN N-terminal kinase (JNK), respectively [22]. The Wnt/PCP signaling directs the cell polarity to induce cell asymmetry, which is crucial for various developmental processes. In the Wnt/Ca^2+^ pathway, Wnt ligand–FZD formation activates phospholipase C (PLC), which releases Ca^2+^ from the intracellular stores. The raised levels of Ca^2+^ induce the activation of protein kinase C, protein kinase II, and calcineurin [23,24]. Although many studies have been conducted in β-catenin-independent signaling, new Wnt ligands, unknown modulators, and cytosolic signaling mediators are constantly being discovered. Thus, the whole map of non-canonical Wnt signaling is still unclear and is different in each biological context.

## 2. Alterations of Wnt Signaling in Cancers

Aberrant Wnt signaling has been implicated in the tumorigenesis of various cancers (Figure 1). For example, genetic mutations of Wnt signaling in colorectal cancer (CRC) are observed in more than 90% of patients and are well-known cancer-driving alterations [25,26]. Genetic alterations in Wnt signaling primarily occur through gene mutation in *APC*, *ZNRF3*, *CTNNB1* (encodes β-catenin), *AXIN1*, and *RSPOs* [3]. In addition, overexpression, downregulation, and copy number changes have also been observed in the genes of Wnt signaling. These genetic alterations of patterns, frequency, and affected genes vary by cancer type [26] (Figure 1). Notably, in most cancers, alterations in Wnt signaling are mainly related to the direction of Wnt signaling activation. Additionally, importantly, the suppression of the hyperactivation of Wnt signaling has shown excellent tumor suppression effects in many preclinical models.

### 2.1. Driver Alterations of Wnt Signaling in Cancers

In colorectal cancer (CRC), the loss-of-mutation of *APC* and the activating mutation of *CTNNB1* are frequently observed [26,27]. The mutation frequency of *APC* is about 60~80%, and that of *CTNNB1* is about 5~10% [25]. The mutation of *APC* and *CTNNB1* leads to excessive activation of Wnt signaling by inducing the stabilization of β-catenin, which initiates intestinal tumors [26]. In addition, about 8~13% mutations of *RNF43* were found in CRC patients [28]. Mutations in *ZNRF3/RNF43* render cancer hypersensitive to Wnt ligands, leading to the hyperactivation of Wnt signaling and promotion of tumorigenesis [28].

In gastric cancer (GC), 13~22% *APC* mutations are detected [4,27]. *APC* mutation is sufficient to initiate gastric tumors in in vivo models [27]. Thus, the driver role of Wnt signaling mutations is demonstrated. Mutations in *RNF43*, another Wnt signaling component, were observed in 4.3% of the MSS (microsatellite stable) subtype (~20% of GC patients) and 54.6% of the MSI (microsatellite instable) subtype (~80% of patients) and were considered key driver mutations [29,30]. In addition to mutations in Wnt signaling, the overexpression of *Wnt1* and *Wnt6* has been observed in gastric cancer [31,32]. The overexpression of *Wnt1* or a combination of prostaglandin pathways induces preneoplastic lesions or invasive gastric cancer, respectively, in mouse models [33]. Indeed, Wnt signaling activations were also observed in two major predisposed causes of gastric cancer development, *CDH1* mutation (encodes E-cadherin) and Helicobacter pylori infections [34,35,36]. These results imply that Wnt signaling is involved both in tumor-driving and -promoting roles in gastric cancers.

In hepatocellular carcinoma (HCC), almost 50% of mutations in the Wnt signaling pathway are activation mutations [37]. Approximately 20~25% of *CTNNB1*, encoding β-catenin, ~10% of *AXIN1*, and ~3% of *AXIN2,* are mutated [37,38]. Additionally, FZD3, FZD6, and FZD7 receptors, Wnt 3, Wnt4, Wnt 5A ligands, and the modulator RSPO2 are frequently overexpressed or amplified in HCC [39]. These genetic alterations in Wnt signaling are closely associated with the progression of hepatocarcinoma (HCC). Although the activating mutation of *CTNNB1*, the most frequent mutation in HCC, does not induce a tumor, the loss-of-*APC* or *Axin1* mutation, both less common in HCC, are able to do so [40,41].

### 2.2. Cancer-Promoting Alterations of Wnt Signaling in Cancers

In addition to the oncogenic role of Wnt signaling, many genetic alterations in the Wnt signaling of specific types of cancers do not drive tumorigenesis but play a tumor-promoting role during cancer progression [39,42]. Additionally, Wnt signaling supporting tumor niches is essential for promoting various primary and metastatic cancers [43,44].

In lung cancer, the mutation of *CTNNB1* and *APC* genes is not common [45,46]. However, the hyperactivation of Wnt signaling is associated with tumor formation, relapse, and poor prognosis [35,47]. In small-cell lung cancer (SCLC), the mutation in *APC* and *CHD8*, which inhibits transcription mediated by *CTNNB1*, is related to the relapse of SCLC [48]. In non-small cell lung cancer (NSCLC), Wnt ligands (Wnt1, Wnt2, Wnt3, and Wnt5a) and other signaling modules (FZD8, PORCN, and TCF-4) are overexpressed [49]. A recent elegant study shows that the Wnt-producing niche is essential for lung adenocarcinoma tumorigenesis [38]. Moreover, this study shows that the pharmacological inhibition of the Wnt niche by the PORCN inhibitor LGK974 suppresses lung tumor growth in in vivo mouse models.

Pancreatic ductal acinar cell carcinoma (PDAC) exhibits approximately 4~10% mutations in *RNF43* and <1% mutations in *APC* and *CTNNB1* [27]. Although genetic alterations in Wnt signaling are rarely detected in PDAC, many studies have suggested that the activation of Wnt signaling is involved in pancreatic tumorigenesis [42,50]. Indeed, the inhibition of the interaction between FZD and Wnt ligands prevents tumorigenesis. In addition, in patient-derived PDAC cell lines harboring RNF43 mutations, the inhibition of porcupine and FZD5 suppresses the growth of PDAC cells [42,51].

## 3. Targeting Strategies for Wnt Signaling in Cancer

Since aberrant Wnt activation initiates or promotes tumorigenesis, Wnt-targeting strategies in cancer therapy are directed toward the downregulation or restoration of overactivated Wnt signaling. Current Wnt targeting strategies can be classified into four categories according to the target location in Wnt signaling: targeting Wnt ligands, targeting Wnt receptors, targeting destruction complex, and targeting β-catenin/transcriptional factors. Extracellular Wnt ligands and receptors are good targets for specific antibodies. However, intracellular signaling components, such as the destruction complex and β-catenin/transcriptional factors, cannot be targeted with antibodies. Only small chemicals and peptides are available to target those. Thus, the enzymatic domains of Wnt signaling components are considered key targets of intracellular components.

In this section, we describe current Wnt targeting strategies as well as recent updates, with meaningful clinical trials involving each strategy (Table 1 and Table 2, and Figure 2).

### 3.1. Targeting Wnt Ligands

Targeting cancer-specific Wnt ligands in certain cancers may be an excellent way to increase the specificity of cancer treatment. However, Wnt ligands often act redundantly, and which Wnt ligands are essential is not fully understood in each cancer context [5]. Additionally, in most cases, targeting Wnt ligands may be less helpful. This is because mutations in Wnt signaling mainly occur downstream of Wnt ligands [27]. Therefore, targeting Wnt ligands is likely more beneficial in cases where Wnt acts as a component of a cancer-propagating niche.

Indeed, the PORCN inhibitor LGK974 (also referred to as WNT974) prevents the secretion of Wnt by inhibiting the palmitoylation of Wnt and shows dramatic suppression of lung tumorigenesis and metastatic CRC progression [43,54]. In metastatic colorectal cancer, WNT974 with a combination of LGX818 and cetuximab was used in a clinical trial (NCT02278133) (Table 1 and Figure 2). Other PORCN inhibitors, RXC004 (NCT03447470, NCT04907539, and NCT04907851) and ETC-159 (NCT02521844) have also been used in clinical trials [57,59] (Table 1 and Figure 2). RXC004 caused the suppression of tumor growth in pancreatic cancer xenograft and gastric cancer PDX models [57]. ETC-159 inhibits the secretion and activity of all Wnt ligands [59].

Soluble Wnt modulators could also be a good target for Wnt-targeted therapy. Soluble FZD-related proteins (SFRP) restrain the Wnt signaling pathway. SFRP has a cysteine-rich domain (CRD) homologous to FZD-CRD. This domain acts as an SFRP to bind competitively to Wnt ligands in order to inhibit Wnt signaling [129,130]. V3Nter, an SFRP-derived polypeptide, binds to Wnt3A and inhibits Wnt signaling in CRC [129]. However, Wnt-targeted therapeutics using SFRP have not been tested in clinical trials.

The targeting of another soluble Wnt modulator, dickkopf-related protein family (DKK), has proved to be more promising in clinical trials. DKKs are composed of five types, of which dickkopf-1, DKK1, binds to LRP5/6 and was initially considered to inhibit canonical Wnt signaling [4,19,131]. However, several studies have since reported that DKK1 is also related to activating the non-canonical Wnt pathway [132,133,134,135]. Although the mechanism of DKK1 in regulating Wnt signaling is unclear, importantly, DKK1 overexpression has been observed in many types of cancer [136], and the inhibition of DKK1 suppresses tumorigenesis in several cancers [137,138,139,140]. Therefore, DKK1 is emerging as an important target for cancer therapy. DKN-01 is a humanized antibody that binds to DKK1, inhibiting cancer progression [68]. Phase 1 and 2 clinical trials are currently underway for monotherapy and combination therapy of DKN-01 in various cancer types, such as colorectal, gastric, endometrial, and liver cancer (NCT05480306, NCT04057365, NCT03395080, and NCT03645980) (Table 1 and Figure 2). As many clinical trials for DKN-01 are underway, we can hopefully expect the first Wnt-targeted therapy targeting DKK1 to receive FDA (Food and Drug Administration) approval soon.

### 3.2. Targeting Wnt Receptors

Targeting Wnt receptors is challenging in cancers harboring mutations downstream of Wnt signaling components. However, cell surface receptors are relatively easy to develop antibodies for, making them attractive drug targets. Additionally, antibodies targeting cancer-specific Wnt receptors could give more specificity with fewer side effects [141]. Furthermore, Wnt receptor-specific antibodies can be applied to Wnt niches in monotherapy and combination therapy with existing anticancer drugs. Here, we introduce cases using receptor-specific antibodies or clinical trials that may be more feasible for therapeutic application (Table 1).

A fair number of FZD-targeting antibodies are currently undergoing clinical trials (Table 1). A monoclonal antibody (mAB), OMP-18R5, binds to five FZD receptors (FZD1, FZD2, FZD5, FZD7, and FZD8) and blocks the canonical Wnt pathway [65]. This OMP-18R5 mAB inhibits tumor growth in colon, breast, and pancreatic cancer cells [65]. OMP-18R5 is being tested in a phase 1 clinical trial (Table 1 and Figure 2). OTSA-101 is a mAB targeting FZD10 and is in a phase 1 clinical trial for sarcoma [66] (Table 1 and Figure 2). An Fc fusion protein, OMP-54F28, has an extracellular N-terminal cysteine-rich domain (CRD) of FZD8 [60]. By acting as a competitor of FZD8, OMP-54F28 inhibits Wnt ligand signaling. Testing OMP-54F28 is in a phase 1 clinical trial for various solid tumors, including liver, ovarian, and pancreatic cancers (Table 1 and Figure 2).

Naturally secreted R-spondins induce the accumulation of FZD receptors and enhance Wnt signaling. Thus, inhibiting natural or cancer-specific RSPOs could be a therapeutic target for Wnt signaling. OMP-131R10 (Rosmantuzumab) is a monoclonal antibody targeting R-spondin 3 (RSPO3) that has now completed a phase 1 clinical trial in advanced solid or relapsed tumors (NCT02482441) [52] (Table 1 and Figure 2).

### 3.3. Targeting the β-Catenin Destruction Complex

The β-catenin destruction complex, consisting of APC, CK1, AXIN, and GSK3β, plays a vital role in controlling the concentration of cytosol and nuclear β-catenin [142]. Loss-of-function mutations in the destruction complex components are found in multiple cancers [27]. Thus, the restoration of the destruction complex or enhancement of its function is required for cancer therapy [19].

CK1 and GSK3β phosphorylate β-catenin, which subsequently induces the ubiquitination and proteasomal degradation of β-catenin [4]. Thus, activating CK1 and GSK3β could enhance the function of the destruction complex. Pyrvinium is an FDA-approved drug used initially as an effective anthelmintic against pinworms [143]. Later, people found that Pyrvinium inhibits Wnt signaling by enhancing CK1 kinase activity [74,144] (Table 1 and Figure 2). SSTC3 is a preclinical small-molecule activator of CK1, which has better pharmacokinetics than existing CK1-activating drugs [106]. CCT031374 is a preclinical activator of GSK3β that may act as a Wnt signaling inhibitor by inducing the degradation of wild-type β-catenin [107] (Table 2).

Sulindac is an FDA-approved non-steroidal anti-inflammatory drug [145]. Later, Sulindac exhibited inhibition of the Wnt/β-catenin pathway by binding to the PDZ domain of Dvl, blocking the destruction complex at the cell membrane [73]. Other small chemicals bind to the PDZ domain of Dvl and are also undergoing preclinical tests [99,100,101,102,103,104,105] (Table 2).

Another way to enhance the destruction of complex-mediated β-catenin degradation is to stabilize the AXIN. Tankyrase belongs to the poly ADP-ribose polymerase family and induces the proteasomal degradation of AXIN [146,147]. Thus, tankyrase inhibition significantly increases the stability of the AXIN protein [147]. Various tankyrase inhibitors, including IWR-1, G007-LK, JW55, JW74, and XAV939, have been evaluated in preclinical studies [117,118,119,120] (Table 2). E7449, an inhibitor of Poly (ADP-ribose) Polymerases (PARP) 1/2 and tankyrase (TNKS) 1/2 is undergoing phase 1 and 2 clinical trials for advanced solid tumors (NCT03878849 and NCT01618136) [148] (Table 1 and Figure 2).

H^+^-ATPase (v-ATPase) mediates the vesicular acidification of lysosomes. This acidification of lysosomes is required for the lysosomal degradation of Wnt receptors/co-receptors and APC, which hyperactivates Wnt/β-catenin signaling [148,149]. Thus, the treatment of v-ATPase inhibitors led to the suppression of Wnt/β-catenin signaling and tumorigenesis in several contexts [148]. A substantial number of v-ATPase inhibitors are undergoing phase 2 clinical trials (Table 1 and Figure 2).

Targeting the intracellular protein complex is quite challenging. Moreover, since most Wnt-associated cancers carry *APC* mutations, enhancing the destruction complex’s function is impossible. In addition, enhancing destruction complexes via v-ATPase inhibitors can lead to a broad range of potential side effects induced by pan-lysosomal inhibition.

### 3.4. Targeting β-Catenin and Its Transcription Partners

β-catenin acts as a central transcriptional activator by forming a complex with its transcriptional partner, TCF/LEF. β-catenin also recruits co-activators such as cAMP response element binding protein (CBP) and p300 [20,21]. The interaction of these factors is essential for the full activation of β-catenin transcriptional responses. Thus, blocking these interactions could be a strategy for inhibiting Wnt signaling.

ICG-001 and its analog PRI-724 interfere with the interaction between β-catenin and CBP [80,81]. Indeed, PRI-724 blocked β-catenin/CBP interaction in a liver fibrosis model [81]. The testing of PRI-724 in advanced solid cancers is currently ongoing and phase 1 clinical trials have been completed (Table 1 and Figure 2). The stabilized α helix of BCL9 (SAH-BCL9) hinders the interaction between β-catenin and BCL9, a transcriptional coactivator [126]. Vitamin D3 (1α,25-dehydroxyvitamin D3) also disrupts the interaction between β-catenin and TCF-4. The vitamin D receptor binds its ligand and competes with TCF-4 to bind the β-catenin [150]. Several clinical trials using vitamin D3 as a supplement are underway in phase I–III clinical trials combined with standard chemotherapy in multiple cancers (NCT02726113, NCT04166253, NCT02603757, and NCT01150877) (Table 1 and Figure 2).

Theoretically, interfering with the binding of β-catenin to its transcriptional partner might be an excellent strategy. It would also be ideal if we could specifically inhibit the mutant beta-catenin. However, it is tremendously difficult to understand the specific interactions between proteins. Moreover, finding chemicals that prevent such interactions is another obstacle. Although there are many limitations, several small-molecule compounds are undergoing pre- or clinical tests (Table 1 and Table 2), and new small-molecule compounds are continuously emerging.

### 3.5. Current Caveats of Targeting Wnt Signaling

Despite ongoing clinical trials, devastating adverse effects on tissue homeostasis and regeneration occur while targeting Wnt signaling. Therefore, Wnt signaling-targeted therapy holds excellent promise but carries high risks in cancer treatment [151]. In this context, potent inhibition of general Wnt signaling might not be a smart method. Lower doses and combination therapy with other anti-cancer therapies may provide an alternative and rational approach to Wnt-targeted therapies. Additionally, targeting Wnt-supporting cancer niches may be more beneficial than Wnt-targeted monotherapy.

As has been mentioned elsewhere, targeting Wnt ligands and receptors against antibodies can give specificity to cancer therapy and cause fewer side effects compared to chemical drugs [152]. However, it is difficult to identify specific ligands and receptors in each cancer context [141]. Moreover, these strategies cannot be effective for most Wnt-mutated cancers that carry *APC* or *CTNNB1* mutations downstream of Wnt ligands and receptors. Similarly, targeting intracellular destruction complexes is also not very beneficial for these Wnt mutant cancers.

Although targeting mutant or nuclear β-catenin is theoretically ideal, finding small chemical inhibitors that prevent β-catenin interactions is extremely challenging.

## 4. New Targeting Strategies for Wnt Signaling in Cancer

Though many limitations exist in Wnt targeting fields, new trials are continuously ongoing. Moreover, clinical trials are underway for promising Wnt signaling-targeted drugs, such as DKK1 and FZD-targeting antibodies. Thus, hope still exists to develop the first Wnt-targeting drug. Furthermore, new advanced technologies are being developed to turn ‘undruggable’ proteins into ‘druggable’ proteins. Here, we introduce a new wave of Wnt-targeting strategies that makes use of the latest cutting-edge technologies.

### 4.1. PROTAC/Molecular Glue-Based Wnt Signaling Targeting

#### 4.1.1. PROTAC

PROteolysis TArgeting Chimera (PROTAC) is a bifunctional molecule composed of an E3 ligase, a protein of interest (POI), and a linker capable of linking two ligands [153].

The E3 ligase ubiquitinates POIs that interact with PROTAC, and the target protein is eventually degraded by the proteasome [154]. PROTAC was first proposed in 2003. However, little progress was made for a long time due to the limited number of binders and poor intracellular delivery [154]. In recent years, PROTAC has started to make remarkable progress thanks to various cutting-edge technologies, such as new ligands that bind with E3 ligases, advanced linker technology, and accumulated chemical design knowledge [154]. In particular, the discovery of small ligands for the E3 ligase cereblon (CRBN) and Von Hippel–Lindau disease (VHL) provided great insights into the design of PROTAC and revolutionized the PROTAC field [153,154,155]. As a result, PROTAC is currently attracting significant attention and being actively studied as an efficient method of erasing target proteins.

In Wnt-targeted cancer therapy using PROTAC, β-catenin is an attractive intracellular target. xStAx-VHL is a PROTAC-targeting β-catenin consisting of xStAx and VHLL [153] (Figure 3 and Table 3). xStAx is a peptide that shows high similarity with the β-catenin-binding domain of AXIN. xStAx-VHL induces dose-dependent and lasting degradation of β-catenin in cell lines and APC^−/−^ organoids. Additionally, xStAx-VHL inhibits the growth of CRC xenografts and APC ^Min/+^-driven intestinal tumors [153].

Since PROTAC does not work based on equilibrium occupancy, the working doses of PROTAC are very low, i.e., nanomolar concentrations [155]. Thus, PROTACs are expected to have low toxicity and high selectivity compared to conventional inhibitors. For example, the BCL-XL-specific PROTAC DT2216 shows low toxicity in vitro and inhibits the growth of xenograft tumors [162]. Additionally, DT2216 shows target specificity to BCl-XL over all other BCL-2 family members (BCL-2, BCl-XL, MCL-1) [162].

#### 4.1.2. Molecular Glue

A small molecule that induces or stabilizes the neo-interaction between two different proteins is called a ‘molecular glue.’ Some molecular glues, such as PROTAC, promote targets’ degradation by inducing new interactions between E3 ligase and target proteins. Additionally, molecular glue could complement PROTACs with far more advantages in terms of molecular weight than PROTACs [156]. In this context, a recent study reports β-catenin’s molecular glue. Usually, mutated β-catenin cannot bind to β-TrCP, a natural E3 ligase. However, β-catenin molecular glue NRX-252114 restores the interaction and induces the proteasome degradation of mutated β-catenin [156] (Figure 3 and Table 3).

#### 4.1.3. Other Protein Degradation Technologies

Recently, various protein degradation technologies, such as PROTAB (proteolytic-targeting antibody), AUTOTAC (AUTOphagy-TArgeting Chimera), and LYTAC (lysosome-targeting chimeras), have been developed [163,164]. These new platforms could be used in Wnt-targeting strategies for cancer treatment in the near future.

Indeed, a recent study showed a new targeting strategy, PROTAB, that uses Wnt signaling components. PROTAB is an antibody that induces the proteolysis of extracellular target receptors by linking to transmembrane E3 ligase ZNRF3, a negative regulator of Wnt signaling [165]. A ZNRF3-HER2 PROTAB induces the degradation of HER2 in CRC cells (SW48) and tumors of the SW48 xenograft model [165]. Additionally, ZNRF3-HER2 PROTAB shows tumor-specific degradation in CRC organoids. Though this platform does not directly target Wnt signaling compartments, it might be suitable for targeting the membrane receptors of the Wnt pathway.

### 4.2. Antibody–Drug Conjugate (ADC)-Based Wnt Signaling Targeting

An antibody–drug conjugate (ADC) consists of a monoclonal antibody (mAB), a cytotoxic drug (payload or warhead), and a chemical linker [166]. ADCs induce the apoptosis of antibody-bounded antigen-expressing cells. ADCs may enable more precise and effective targeting and elimination of target cells, combining the advantages of monoclonal antibodies and cytotoxic drugs [159].

Septuximab vedotin (F7-ADC) is an ADC for targeting FZD7 [157] (Figure 3 and Table 3). It comprises a human FZD7 antibody and the microtubule-inhibiting drug monomethyl auristatin E (MMAE). Ovarian serous cystadenocarcinoma overexpresses the Wnt receptor FZD7 [157]. Thus, FZD7 could be a tumor-specific antigen in ovarian serous cystadenocarcinoma. Indeed, Septuximab vedotin (F7-ADC) kills ovarian cancer cells without toxicity in vitro and in vivo.

PTK7, involving both Wnt and VEGF signaling, is associated with cancer drive and repression [158]. In the Wnt pathway, PTK7 is a co-receptor of Wnt ligands. WNT2A binds to the dimer of FZD7 and PTK7. The interaction of WNT2A, FZD7, and PTK7 inhibits the canonical Wnt pathway, while the interaction of WNT5A, ROR2, and PTK7 activates Wnt/PCP pathway. Targeting PTK7 could be suitable for specifically targeting Wnt responder CSCs in cancers. PF-06647020 is a PTK7-targeted ADC [158] (Figure 3 and Table 3). PF-06647020 delivered anti-cancer drugs effectively and has shown anti-cancer effects in various cancer cell lines and PDXs (NSCLC, OVAC, and TNBC). A phase 1 clinical trial of PF-06647020 has been completed in advanced solid tumors.

LGR5-targeted ADCs, LGR5–MC-vc-PAB–MMAE and LGR5–NMS818, were recently developed. These LGR5-targeted ADCs target the LGR5-expressing population of tumor-initiating cells or cancer stem cells (CSCs) [159] (Figure 3 and Table 3). These cells are highly responsive to Wnt signaling and are essential for tumor progression [167,168,169]. Although these ADCs do not directly inhibit the Wnt pathway, they kill Wnt responder cells such as CSCs. Moreover, these LGR5-targeted ADCs can dramatically restrain tumor growth and recurrence in vivo.

Currently, there are still difficulties in applying ADCs to cancer treatment. Most approved ADCs have side effects such as hematotoxicity [170]. Moreover, they have heavier molecular weights than conventional cytotoxic drugs, which causes poor delivery efficiency of drugs to tumors [170]. Despite these limitations, ADCs have substantial potential and are of great interest as an excellent method of targeting tumors specifically.

### 4.3. Oligonucleotide-Based Wnt Signaling Targeting

Oligonucleotides are emerging therapeutics that use small nucleotides (15–30 bp) with chemical modifications mainly to regulate the target gene’s transcription [171]. These small oligonucleotide therapeutics come in various forms, such as MicroRNAs (miRNA), antisense oligonucleotides (ASOs), and short-interfering RNAs (siRNAs). In addition, numerous new types of oligonucleotides, modifications, and delivery systems are continuously being developed. Thus, oligonucleotide therapeutics are expected to be applicable to cancer treatment soon, which may involve targeting Wnt signaling.

#### 4.3.1. MicroRNAs and siRNAs

miRNAs and siRNAs consist of short sequences to target RNA through partial base pairing [172]. In prostate cancer, miR-15a and miR-16-1 are reported as tumor suppressors and downregulate Wnt3a in prostate cancer cells [173]. TargomiRs are minicells that contain miR-16-based mimic microRNA, which has 23 base pairs, and target EGFR and Wnt3a on the cell surface [174]. TargomiRs have completed clinical trials (Phase 1, NCT02369198) for the treatment of recurrent malignant pleural mesothelioma and non-small cell lung cancer [174]. Myc is a central effector of the β-catenin-dependent Wnt pathway [175]. DCR-Myc is an anti-Myc DsiRNA (Dicer-substrate small interfering RNA) that inhibits tumor growth in vivo [176]. These drugs recently completed clinical trials in hepatocellular carcinoma (Phase 1b/2, NCT02314052) and solid tumors (Phase 1, NCT02110563). Though these miRNAs and siRNAs have been tested in clinics, oligonucleotide therapies have potential hurdles, off-target effects, and inefficient delivery [177]. Additionally, these miRNAs and siRNAs do not directly target Wnt-signaling compartments.

#### 4.3.2. Antisense Oligonucleotides

Antisense oligonucleotides (ASOs) are single-stranded DNA analogs that consist of 16–22 bases [160]. ASO binds to the target RNA sequence and controls its expression via various mechanisms (Figure 3). Recently, ribose substitutions such as 2′-O-methoxyethyl (2′-MOE) and locked nucleic acid (LNA) have improved the stability and accuracy of target binding [178]. Importantly, various antisense oligonucleotides (ASO) have recently gained FDA approval [171,179,180,181].

Several ASOs that target the Wnt pathway indirectly or directly are undergoing preclinical studies. LNA-modified ASO is reported as an indirect targeting ASO for the Wnt pathway (Figure 3 and Table 3). The long non-coding RNA (lncRNA) AC10401.1 is highly expressed in head and neck squamous cell carcinoma (HNSCC) and is related to poor prognosis in HNSCC patients. It acts as a competing endogenous RNA to miR-6817-3d in the cytoplasm and increases the stability of Wnt2B, which activates the canonical Wnt pathway [160]. Furthermore, a combination of LNA-modified ASO and a Wnt signaling inhibitor restrain AC10401.1 and inhibit cancer cell growth in cell-line and patient-derived xenograft models. This result implies that LNA-ASO can potentially suppress tumorigenesis by targeting Wnt signaling.

β-catenin is directly targeted by 2’-O-methoxyethyl chimeric ASO. Treatment with this β-catenin-targeted ASO showed a specific decrease in β-catenin expression in liver and white adipose tissue in high-fat-fed C57BL/6 mice [161] (Figure 3 and Table 3). These data show that the specific targeting of β-catenin is feasible in in vivo models. Although ASOs directly targeting Wnt signaling have not been tested, these results will soon emerge in preclinical and clinical cancer models.

## 5. Conclusions and Future Perspectives

Despite the contribution of Wnt signaling to tumorigenesis being obvious and Wnt signaling inhibition having shown significant effects in preclinical models, no Wnt signaling-targeted drugs have proved clinically successful in cancer or other diseases [4,6]. The main reason may be that Wnt signaling is responsible for a broad range of physiological regulations [8]. This feature could trigger diverse side effects as a consequence of treatment. Additionally, it may be because Wnt-targeted therapy itself does not have significant benefits over conventional anticancer drugs. Thus, recent Wnt targeting methods have focused on increasing treatment specificity, minimizing side effects, and combination therapies that make up for the low effectiveness of safe-dose Wnt-targeting drugs.

In addition to these efforts, combining the newest targeting strategies, such as PROTAC/molecular glue, ASOs, and ADCs, was recently introduced to Wnt-targeted therapy. Adopting these technologies and accumulating knowledge on mechanisms of action from past and current Wnt-targeting trials have allowed for a new wave of Wnt signaling-targeting therapies.

These new targeting technologies could enhance the specificity of drugs. PROTACs eliminate their targets; they do not inhibit them. ADC combines the advantages of antibodies and cytotoxic drugs, ameliorating conventional side effects. The design of ASOs is straightforward and assigns specificity toward targets. These three methods can help improve the limitations of existing targeting strategies. Additionally, numerous combinations of these new techniques are possible.

ASOs were first suggested in the early 2000s. However, the initial ASO system had limitations: poor cellular delivery, not being sufficiently stable under degradation by nuclease, and off-target effects. Additionally, some modified ASOs show high toxicity [182]. Overcoming these limitations requires time and the use of ASOs as therapeutic drugs. Nusinersen, an ASO drug, was approved in 2016 [183]. This drug has been enormously successful for spinal muscular atrophy patients. Additionally, many FDA-approved ASO drugs are used to treat rare diseases [179,180,181]. In cancer, two ASOs, danvatirsen and AZD5312 (NCT02144051), are currently undergoing clinical trials [184,185]. Danvatirsen degrades STAT3 mRNA via RNase H1 and AZD5312 degrades androgen receptor mRNA.

In conclusion, before the advent of third-generation ASOs and FDA-approved ASOs, it was assumed that ASOs could not be used for treatment. Twenty years ago, before the success of imatinib, the first FDA-approved kinase drug [186], people thought kinase drugs were impossible. Likewise, prior to Herceptin, antibodies were also considered unavailable for therapeutic use [187,188]. Therefore, the first success in Wnt-targeted therapeutics will be a significant milestone. Based on recent advanced technologies, PROTAC/molecular glue, ASO, and ADC, and accumulated knowledge of Wnt signaling, we cautiously expect that the first case of a new cancer treatment targeting Wnt signaling will appear in the near future.

## Figures and Tables

**Figure 1 cells-12-01110-f001:**
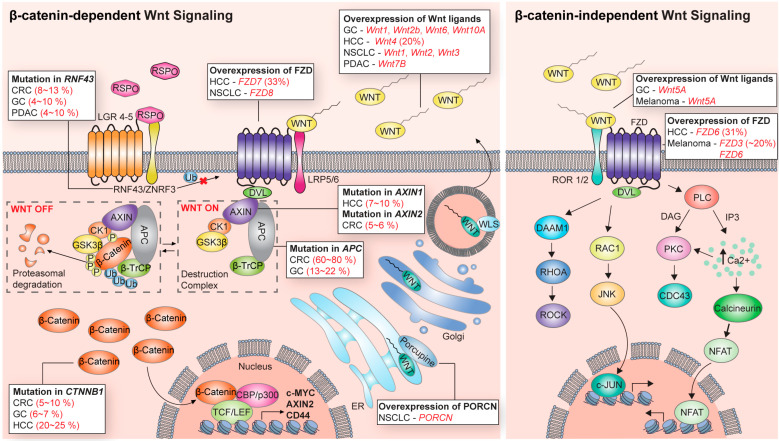
Wnt signaling and its alteration in various cancers. Signal transduction of β-catenin-dependent and independent Wnt signaling. The frequencies of genetic alteration of individual Wnt signaling modules in indicated cancers were summarized.

**Figure 2 cells-12-01110-f002:**
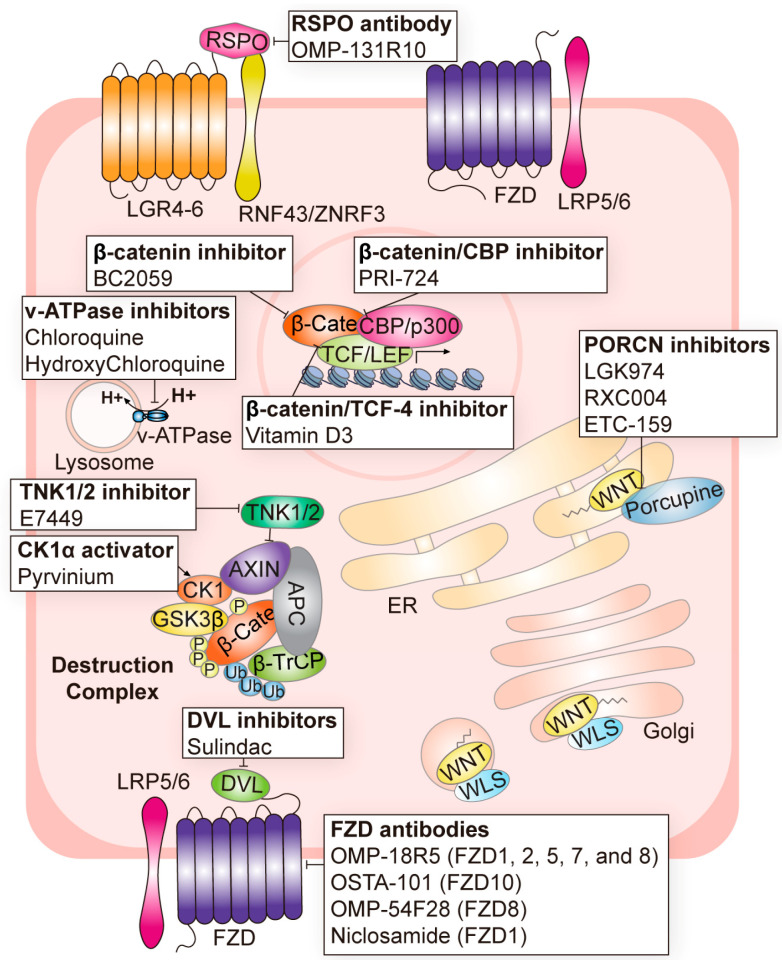
Wnt signaling modulating agents in clinical trials for cancer treatment. Promising clinical agents targeting the Wnt signaling module were displayed along with their targets.

**Figure 3 cells-12-01110-f003:**
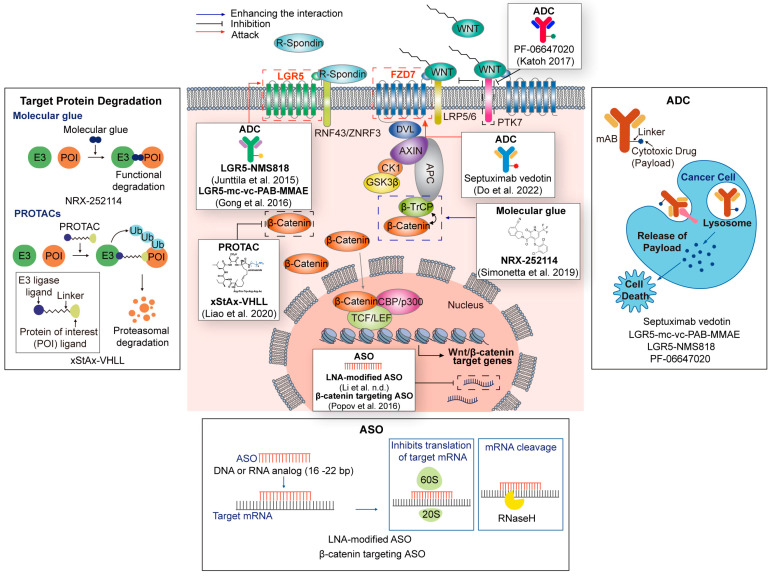
Targeting Wnt signaling pathway by the latest technologies. β-catenin targeted PROTACs (xStAx-VHL) recruits E3 ligase to β-catenin, inducing the degradation of β-catenin using the ubiquitin–proteasome system. Additionally, the molecular glue of β-catenin and its E3 ligase β-TrCP, NRX-252114, enhances the β-catenin degradation. ADCs (LGR5-mc-vc-PAB-MMAE, LRG5-NMs818, and PF-06647020) targeting LGR5- or PTK7-expressing Wnt responder cells bind to the target cells and release a cytotoxic drug to induce the cell death. ASO (LNA-modified ASO, β-catenin targeting ASO) binds to the target region of RNA and suppresses expression.

**Table 1 cells-12-01110-t001:** Wnt signaling targeting agents in clinical trials for cancer treatment.

Components Name	Target	Cancer	Clinical Phase	Inhibition of Canonical or Non-Canonical Wnt Signaling	Refs.
OMP-131R10	RSPO	Advance Relapsed Tumors (CRC)	Phase 1 (NCT02482441)	Canonical	[52]
Foxy-5	WNT5A mimic	Colon Cancer Metastatic Breast and Colon/Prostate Cancer Metastatic Breast, Colon, and Prostate Cancer	Phase 2 (NCT03883802) Phase 1 (NCT02020291) Phase 1 (NCT02655952)	Non-canonical	[53]
LGK974	PORCN	Metastatic Colorectal Cancer Metastatic Head and Neck Squamous Cell Carcinoma Solid Malignancies	Phase 1/2 (NCT02278133) Phase 2 (NCT02649530) Phase 1 (NCT01351103)	Canonical/Non-canonical	[3,43,54]
RXC004	PORCN	Colorectal Cancer Solid Tumor Advanced Solid Tumors	Phase 2 (NCT04907539) Phase 1 (NCT03447470) Phase 2 (NCT04907851)	Canonical/Non-canonical	[55,56,57]
ETC-159	PORCN	Solid Tumors	Phase 1 (NCT02521844)	Canonical/Non-canonical	[58,59]
OMP-54F28	FZD8	Hepatocellular Cancer Ovarian Cancer Pancreatic Cancer/Solid Tumors	Phase 1 (NCT02069145) Phase 1 (NCT02092363) Phase 1 (NCT02050178) Phase 1 (NCT01608867)	Canonical	[60,61,62]
Niclosamide	FZD1	Colon Cancer Metastatic Prostate-Carcinoma Acute Myeloid Leukemia	Phase 1 (NCT02687009) Phase 1 (NCT03123978) Phase 1 (NCT02532114) Phase 2 (NCT02807805) Phase 1 (NCT05188170) FDA-approved antihelminth	Canonical	[63]
OMP-18R5	FZD1/2/5/7/8	Solid Tumors Pancreatic Cancer Metastatic Breast Cancer	Phase 1 (NCT01345201) Phase 1 (NCT01957007) Phase 1 (NCT02005315) Phase 1 (NCT01973309)	Canonical	[64,65]
OTSA-101	FZD10	Sarcoma	Phase 1 (NCT01469975)	Canonical	[66]
BNC101	LGR5	Colorectal Cancer	Phase 1 (NCT02726334)	Canonical/Non-canonical	[67]
DKN-01	DKK1	Multiple Myeloma Advanced Solid Tumors/Relapsed NSCLC Relapsed Esophagogastric Malignancies Cholangiocarcinoma Epithelial Endometrial-/Epithelial Ovarian Cancer	Phase 1 (NCT01457417) Phase 1 (NCT01711671) Phase 1 (NCT02013154) Phase 1 (NCT02375880) Phase 2 (NCT03395080)	Canonical/Non-canonical	[68,69,70,71,72]
Sulindac	DVL	Breast Cancer Colorectal Cancer Lung Cancer	Phase 1 (NCT00245024) Phase 2 (NCT04542135) Phase 2 (NCT01856322) Phase 2 (NCT00062023) Phase 2 (NCT00368927) FDA-approved nonsteroidal anti-inflammatory drug	Canonical	[73]
Pyrvinium	CK1	Pancreatic Cancer	Phase 1 (NCT05055323) FDA-approved antihelminth	Canonical	[74]
E7449	TNK1/2	Advanced Ovarian Cancer Advanced Solid Tumors/ B-cell Malignancies	Phase 2 (NCT03878849) Phase 1 (NCT01618136)	Canonical	[75,76]
BC2059	β-catenin	Desmoid Tumor Metastatic NSCLC Recurrent Leukemia Solid Tumor	Phase 1 (NCT03459469) Phase 1 (NCT04780568) Phase 1 (NCT04874480) Phase 1/2 (NCT04851119)	Canonical	[77,78,79]
PRI-724	β-catenin/CBP	Advanced Solid Tumor Advanced Pancreatic Advanced Myeloid- Malignancies	Phase 1 (NCT01302405) Phase 1 (NCT01764477) Phase 1/2 (NCT01606579)	Canonical	[80,81]
SM08502	CLK	Advanced Solid Tumors Solid Tumor	Phase 1 (NCT05084859) Phase 1 (NCT03355066)	Canonical	[82]
Chloroquine	v-ATPase inhibitor	Pancreatic Cancer Solid Tumors Ductal Carcinoma In Situ Glioma and Cholangiocarcinoma Glioblastoma Multiforme	Phase 1 (NCT01777477) Phase 1 (NCT02071537) Phase 1/2 (NCT01023477) Phase 1/2 (NCT02496741) Phase 3 (NCT00224978)	Canonical	[83]
Hydroxy chloroquine	v-ATPase inhibitor	Colorectal Cancer Prostate Cancer Metastatic pancreatic cancer	Phase 2 (NCT01006369), etc. (total of 30 trials completed)	Canonical	[84,85,86,87,88]

**Table 2 cells-12-01110-t002:** Pre-clinical agents targeting Wnt signaling.

Target	Pre-Clinical Agents	Refs.
PORCN	IWP1, IWP2, IWP3, IWP4, IWP12, IWP L6, C59, GNF-6231, GNF-1331	[31,89,90,91,92]
FZD1	DK-520, DK-419	[93,94]
FZD5	IgG-2919	[51]
FZD7	Fz7-21, RHPD-P1, SRI37892	[95,96,97]
FZD8	1094-0205, 2124-0331, 3235-0367, NSC36784, NSC654259, IgG-2919	[98]
WNT/FZD/LRP complex	Salinomycin	[32]
DVL	BMD4702, 3289-8625, J01-017a, FJ9, KY-02061, KY-02327, NSC668036, Peptide Pen-N3	[99,100,101,102,103,104,105]
CK1	SSTC3, CCT031374	[106,107]
GSK3β mimic	TCS 183	[108]
TNKS	XAV939, AZ1366, G007-LK, MSC2504877, G244-LM, IWR-1, JW74, JW55, K-756, NVP-TNKS656, MN-64, RK-287107, WIKI4	[109,110,111,112,113,114,115,116,117,118,119,120]
β-catenin	KY1220, KYA1797K, MSAB,	[78,121]
β-catenin/TCF	PKF115-584, CGP049090, AV-65, PNU-74654	[122,123]
β-catenin/EP300	Windorphen, IQ-1	[124,125]
β-catenin/BCL9	PNPB-29, ZW4864, SAH-BCL9, Carnosic acid	[126,127,128]

**Table 3 cells-12-01110-t003:** Wnt signaling targeting agents using the latest technologies (PROTACs, ADC, and ASO).

Technologies	Name	Target	Refs.
PROTACs	xStAx-VHL	β-catenin	[153]
Molecular glue	NRX-252114	β-catenin/ β-TrCP	[156]
ADC	Septuximab vedotin	FZD7	[157]
	PF-06647020	PTK7	[158]
	LGR5-mc-vc-PAB-MMAE	LGR5	[159]
	LGR5-NMS818	LGR5	
ASO	LNA-modified ASO	AC104041.1 (lncRNA)	[160]
	β-catenin targeting ASO	β-catenin	[161]

## Data Availability

Not applicable.

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
