# Peer review of "A New Wave of Targeting ‘Undruggable’ Wnt Signaling for Cancer Therapy: Challenges and Opportunities"

_cells, 2023, doi:10.3390/cells12081110_

Round 1

Reviewer 1 Report

I am expecting a new edited version of the manuscript. Extensive grammatical editing is required, as well as having colleagues read and correct errornous sentences.

Author Response

We appreciate Reviewer 1’s comments. We fully agree with the comments and sincerely apologize for the incomplete English use. As you suggested, we edited our manuscript's English through an editing service (Certificate is included) and checked by colleagues. And now, we are submitting the improved revised manuscript that meets the publication quality.

Again, thanks for your comments.

Reviewer 2 Report

This is a comprehensive review of WNT signaling and its therapeutic targeting for the treatment of cancer. The article would benefit from the inclusion of a figure depicting the non-canonical pathways of the WNT signaling. Also, it would be of interest to indicate which of the therapeutic approaches targeting WNT pathway component would benefit the inhibition of the canonical and/or the non-canonical pathways. Such info may be included in Table 1.  

Reviewer 3 Report

Thank you for this very useful updated review of this field, there are some spelling errors that should be corrected: e.g.

recpetors/co-repceptors - l. 58

Mutations in ZNFR3/ RNZ43 render cancer hypersensitive to Wnt ligands 133

founded (should be found) - 279

working does - should be working doses - 385

bounded (should be bound) - 406

The challenges of existing ADCs exist (rephrase) - 436

ADCs have substantial possibilities as an excellent method to be targeting tumor specifically (rephrase, e.g. ADCs have substantial possibilities to target  tumors specifically. - 439

there are no ASO directly targeting the Wnt pathway in clinical (there is no ASO directly targeting the Wnt pathway in the clinic)- 463

This result imply LNA-ASO (the result implies that LNA-ASO...) - 473

targeting the β-catenin makes leading to a specific knockdown on β-catenin expression (rephrase) - 475

there are no targeted drugs for Wnt signaling successful in a clinic in cancer and other diseases (rephrase)  - 490

The main reason may rely on Wnt signaling is responsible (rephrase) - 491

Reviewer 4 Report

 The content of the review, which focuses on canonical Wnt signalling, is comprehensive but  significant changes are required to ensure the meaning is clear throughout the review along with revision of English. 

The review is really focused on canonical signalling  and this should be made clear from the beginning.

What are the outputs of the non-canonical pathways? in the PCP pathway these are not transcriptional outputs but cytoskeleton regulation. A bit more detail is needed for this section or remove it completely and make it clear that the review is only focused on canonical Wnt signalling.

Section 1- paragraph 2- needs to be made clear that the pathway a Wnt signals through depends on context. There is evidence that most Wnts can signal through all pathways but some Wnts predominantly signal through one pathway.

Round 2

Reviewer 1 Report

I appreciate the major grammatical editing of the manuscript, which was very interesting to read and gives a timely overview of the strategies and challenges for targeting the Wnt pathway. I want to congratulate the authors on their endeavor. I only have a limited number of issues with the current version that should be corrected or addressed.

General remarks :

Make sure to put gene names in italics

Is there any role for Norrin (also able to activate the Wnt/β-catenin pathway) in cancer?

Specific remarks:

Line 32 and Ref. 2. If I am not mistaken it was not the mutation of the Int-1 gene but the ectopic expression mediated by the integration of the pro-virus that drove the breast cancer.

Line 89: Is there more than one β-catenin signaling pathway?

Line 91: β-catenin binds to the cytoplasmic tail of all classical cadherins, not just E-cadherin.

Line 95 : The GSK in the complex is specifically GSK3β

Line 107 : ­complex -> complexes

Line 116 : The acronym RAC1 does not need to be spelled out. Just mention ‘RAC1’, it is a well-known GTPase

Line 140 : ‘… the loss-of-function mutation of APC …’

Line 151 : explain MSS and MSI

Line 156-158 : explain how CDH1 mutations and Helicobacter mutations are linked to Wnt signaling

Line 166 : ‘… the activating mutations of CTNNB1…’

Line 167 : … change to : ‘…, loss-of-function mutations in APC or Axin1, both less common in HCC, are able to do so.’

Line 174 : ‘..tumor niches are essential …’

Line 214 : ‘… cases where Wnt acts as a component of a cancer-propagating niche.’

Line 258-259 : ‘… can be applied to Wnt niches in monotherapy …’

Line 280-281 : ‘Loss-of-function mutations in the destruction complex components are found in multiple cancers’

Line 299 : ‘.. the AXIN protein.’

Line 322 : wrong references. Correct references are 17 and 18. Check carefully all the references.

Line 323 : change to ‘… for the full activation of β-catenin transcriptional responses.’

Line 339 : subsequent -> such

Line 348 : ‘… context, potent inhibition of general Wnt signaling  …’

Line 360 : nucleus -> nuclear

Line 457 : delete ‘properties’

Line 483 : Anti-Myc DsiRNA -> Anti-Myc siRNA(?)

Line 504 : AC10401.1. -> AC10401.1 (delete final dot)
